# Mixed Neuroendocrine Non-Neuroendocrine Neoplasms: A Systematic Review of a Controversial and Underestimated Diagnosis

**DOI:** 10.3390/jcm9010273

**Published:** 2020-01-19

**Authors:** Melissa Frizziero, Bipasha Chakrabarty, Bence Nagy, Angela Lamarca, Richard A. Hubner, Juan W. Valle, Mairéad G. McNamara

**Affiliations:** 1Department of Medical Oncology, The Christie NHS Foundation Trust, 550 Wilmslow Road, Manchester M20 4BX, UK; Melissa.Frizziero@christie.nhs.uk (M.F.); Bence.Nagy@christie.nhs.uk (B.N.); Angela.Lamarca@christie.nhs.uk (A.L.); Richard.Hubner@christie.nhs.uk (R.A.H.); Juan.Valle@christie.nhs.uk (J.W.V.); 2Department of Pathology, The Christie NHS Foundation Trust, 550 Wilmslow Road, Manchester M20 4BX, UK; Bipasha.Chakrabarty@christie.nhs.uk; 3Division of Cancer Sciences, University of Manchester, Oxford Road, Manchester M13 9PL, UK

**Keywords:** mixed non-neuroendocrine neuroendocrine neoplasms, MiNENs, mixed adeno-neuroendocrine carcinoma, MANEC, 2017 WHO classification, 2019 WHO classification

## Abstract

Mixed neuroendocrine non-neuroendocrine neoplasms (MiNENs) represent a rare diagnosis of the gastro-entero-pancreatic tract. Evidence from the current literature regarding their epidemiology, biology, and management is of variable quality and conflicting. Based on available data, the MiNEN has an aggressive biological behaviour, mostly driven by its (often high-grade) neuroendocrine component, and a dismal prognosis. In most cases, the non-neuroendocrine component is of adenocarcinoma histology. Due to limitations in diagnostic methods and poor awareness within the scientific community, the incidence of MiNENs may be underestimated. In the absence of data from clinical trials, MiNENs are commonly treated according to the standard of care for pure neuroendocrine carcinomas or adenocarcinomas from the same sites of origin, based on the assumption of a biological similarity to their pure counterparts. However, little is known about the molecular aberrations of MiNENs, and their pathogenesis remains controversial; molecular/genetic studies conducted so far point towards a common monoclonal origin of the two components. In addition, mutations in tumour-associated genes, including *TP53*, *BRAF,* and *KRAS*, and microsatellite instability have emerged as potential drivers of MiNENs. This systematic review (91 full manuscripts or abstracts in English language) summarises the current reported literature on clinical, pathological, survival, and molecular/genetic data on MiNENs.

## 1. Introduction

Epithelial neoplasms displaying a coexistence of a neuroendocrine and non-neuroendocrine histology include a wide spectrum of entities composed of a variable proportion of the two histologies (each representing from 1% to 99% of the tumour mass) and have been described in almost all organs [1,2]. The two components of these mixed neoplasms can exhibit variable morphological features (also depending on the site of origin) as well as degrees of differentiation, and can be combined in different patterns; they can be intimately intermingled within the tumour mass (composite tumours) or they can constitute separate, juxtaposed areas of the tumour mass (collision tumours). In other cases, neuroendocrine and non-neuroendocrine features coexist at a cellular level (amphicrine tumours) [1,3]. Besides their pathological heterogeneity, over the years, mixed neuroendocrine/non-neuroendocrine neoplasms have been assigned a number of different definitions, with some redundant or only partially overlapping (a comprehensive list of the terms used in the literature has been reported by La Rosa et al. [2]), giving rise to a huge inconsistency in published data on these neoplasms.

In 2010, mixed neoplasms from the gastro-entero-pancreatic (GEP) tract containing a neuroendocrine and an exocrine component, each of them present in at least 30% of the tumour mass and being malignant, were classified by the World Health Organisation (WHO) as separate entities and named “mixed adeno-neuroendocrine carcinomas” (MANECs) [4]. The rationale behind the 30% threshold is that a lesser represented component is unlikely to influence the biological behaviour of the whole neoplasm. However, this is an arbitrary threshold and not supported by evidence of its clinical relevance or pathogenic significance [1].

In 2017, the WHO renamed MANECs from the pancreas as “mixed neuroendocrine non-neuroendocrine neoplasms” (MiNENs), where the 30% threshold for each component was maintained, but the term “exocrine” was substituted by the more general term “non-neuroendocrine” to include histological variants that cannot be referred to as exocrine (e.g., squamous or sarcomatoid phenotypes), and the term “carcinoma” was substituted by the term “neoplasm” to recognise the fact that occasionally, one or both components are low-grade malignant [5]. Very recently, the WHO has extended the use of the term to all neoplasms meeting the diagnostic criteria for MiNENs arising from any site within the GEP tract [6]. Compared to “MANECs”, the term “MiNENs” is believed to better address the heterogeneous spectrum of possible combinations between neuroendocrine and non-neuroendocrine elements and the variability of morphologies, which are largely determined by the site of origin [2].

Mixed neuroendocrine non-neuroendocrine neoplasms, as per the 2017–2019 WHO definition [5,6], represent an extremely rare diagnosis. According to the Surveillance of Rare Cancers in Europe registry in 2008, the incidence of MiNENs was below 0.01/100,000 cases per annum, and only 96 people were alive with this diagnosis in the whole continent (http://www.rarecare.eu/). Furthermore, evidence from the literature on MiNENs is almost exclusively derived from case reports and small retrospective series. Due to the rarity of this diagnosis, the limited quality of published data, and the use of inconsistent terminology, the epidemiology, prognosis, and best therapeutic management of patients with MiNEN remains unknown.

Based on available evidence, albeit limited and conflicting, the MiNEN is an aggressive entity with a high-grade neuroendocrine component in the majority of cases, and is associated with poor survival outcomes close to those of pure neuroendocrine carcinomas (NECs) [7]. For these reasons, MiNENs are usually treated similarly to their pure NEC counterpart [8]. Alternatively, when the exocrine component is the preponderant and/or most aggressive histology, some clinicians choose to apply the standard of care for adenocarcinomas (ADCs) from the same site of origin [7]. Both practices are based on principles of histological analogy, but are not supported by evidence from prospective randomised trials.

The pathogenesis of MiNENs represents a matter of open debate amongst pathologists and clinicians. Three main theories have been proposed to date [9]: the first theory suggests that the neuroendocrine and non-neuroendocrine components arise independently, in a synchronous or metachronous manner, from distinct precursor cells and merge; the second postulates that the two components derive from a common pluripotent stem cell progenitor, which acquires biphenotypic differentiation during carcinogenesis; a third theory also assumes a common monoclonal origin of the two components, but hypothesises that the neuroendocrine differentiation develops from an initially non-neuroendocrine cell phenotype, through the progressive accumulation of molecular/genetic aberrations, and not vice versa.

The molecular landscape of MiNENs is also poorly understood. A number of studies have recently attempted to identify the key genetic and epigenetic aberrations underlying MiNENs, with a view to better elucidating how this disease develops, and to explore possible biological similarities between its two components, and with their pure counterparts, as well as to identify potential targets for novel therapeutic approaches [10].

This systematic review outlines the epidemiological, clinical, and pathological characteristics and prognosis of GEP-MiNENs, in addition to the most commonly adopted treatment strategies. This review also focuses on reported genetic and epigenetic data, with a view to providing some insights into the biology and pathogenesis of this rare disease.

## 2. Methods

Biomedical electronic databases, including EMBASE, MEDLINE, and PUBMED, and clinical practice guidelines of the European Neuroendocrine Tumour Society (ENETS), the European Society of Medical Oncology (ESMO), and the National Comprehensive Cancer Network (NCCN) were interrogated for all full manuscripts and conference abstracts written in the English language (at least the abstract), and published between January 2010 (the year of the introduction of the definition of MANEC by WHO [4]) and August 2019, using the following bibliographic search strategy:

“mixed adenoneuroendocrine carcinoma *” or “MANEC *” or “mixed neuroendocrine non-neuroendocrine neoplasm *” or “MiNEN *”.

The Preferred Reporting Items for Systematic Reviews and Meta-Analyses (PRISMA) flow diagram for the selection of the studies is reported in Figure 1.

Studies included were those who met at least one of the following criteria:Provision of clinical-pathological and survival data (at least 6 months of follow-up for patients who were alive at the time of publication) on MiNEN or MANEC.Molecular/genetic findings on MiNEN or MANEC (immunohistochemical data were not included, except when used for the assessment of DNA Mismatch Repair (MMR) protein status).

Descriptive statistical analyses (e.g., frequencies and medians) were conducted on data extracted from individual studies. Survival data of individual patients presented in case reports (CRs) were pulled and median values (and related 95% confidence intervals) were estimated by applying Kaplan–Meier analysis. In retrospective studies (RSs) where survival data were provided for individual patients (rather than for the whole cohort), estimation of median survival outcomes was attempted by applying Kaplan–Meier analysis. Microsoft Excel and SPSS statistics software were used.

## 3. Results

A total of 687 publications were screened. Then, 91 (number of patients (*n*) = 2427 patients) were included [11,12,13,14,15,16,17,18,19,20,21,22,23,24,25,26,27,28,29,30,31,32,33,34,35,36,37,38,39,40,41,42,43,44,45,46,47,48,49,50,51,52,53,54,55,56,57,58,59,60,61,62,63,64,65,66,67,68,69,70,71,72,73,74,75,76,77,78,79,80,81,82,83,84,85,86,87,88,89,90,91,92,93,94,95,96,97,98,99,100,101]; 75 publications were full manuscripts, and 16 were conference abstracts. Fifty-five were case reports (CRs) [11,12,13,14,15,16,17,18,19,20,21,22,23,24,25,26,27,28,29,30,31,32,33,34,35,36,37,38,39,40,41,42,43,44,45,46,47,48,49,50,51,52,53,54,55,56,57,58,59,60,61,62,63,64,65], and 36 were retrospective studies (RSs) [66,67,68,69,70,71,72,73,74,75,76,77,78,79,80,81,82,83,84,85,86,87,88,89,90,91,92,93,94,95,96,97,98,99,100,101]. Eighty-four used the term “MANEC” or “mixed adenoneuroendocrine carcinoma”, and seven used the term “MiNEN” or “mixed neuroendocrine non-neuroendocrine neoplasm”. The number (percentage) of publications per geographical area was as follows; Asia 42 (46.1%), Europe 31 (34.1%), North America 13 (14.3%), South America 2 (2.2%), unknown 3 (3.3%).

### 3.1. Clinical-Pathological Characteristics, Treatment Modalities, and Survival Outcomes

The site of origin of the primary tumour was as follows; appendix 60.3% (*n* = 1463), colon–rectum 14.5% (*n* = 351) (colon 11.2% (*n* = 272), rectum 1.9% (*n* = 45), either colon or rectum 1.4% (*n* = 34)), stomach 6.7% (*n* = 162), oesophagus/oesophagogastric junction (OGJ) 5.9% (*n* = 143), pancreas 3.7% (*n* = 90), biliary tract 1.6% (*n* = 39), small bowel < 1% (*n* = 19), anus < 1% (*n* = 3), unknown primary <1% (*n* = 3), liver < 1% (*n* = 1), and GEP non otherwise specified (n.o.s.) 5.9% (*n* = 144). The remaining nine patients, reported in a study by Apostolidis L. et al. [75], had a MiNEN from outside the GEP tract; data related to these patients could not be selectively extracted and discarded, and therefore were included in the analysis.

Data on gender were provided in 77 studies (*n* = 983) [11,12,13,14,15,16,17,18,19,20,21,22,23,24,25,26,27,28,29,30,31,32,33,34,35,36,37,38,39,40,41,42,43,44,45,46,47,48,49,50,51,52,53,54,55,56,57,58,59,60,61,62,63,64,65,68,69,70,72,73,74,75,79,80,82,84,85,86,88,93,94,95,96,97,99,100,101]; 65.6% (*n* = 645) were male, and 34.4% (*n* = 338) were female. The frequency of the two genders according to the primary tumour site could be explored in 71 studies (*n* = 580) [11,12,13,14,15,16,17,18,19,20,21,22,23,24,25,26,27,28,29,30,31,32,33,34,35,36,37,38,39,40,41,42,43,44,45,46,47,48,49,50,51,52,53,54,55,56,57,58,59,60,61,62,63,64,65,68,70,72,73,74,79,80,82,86,88,93,94,95,97,100,101]. In the majority of subgroups per primary tumour site, the male gender was prevalent; stomach 89.5% (*n* = 68 out of 976), oesophagus/OGJ 86.1% (*n* = 87 out of 101), pancreas 66.7% (*n* = 24 out of 36), colon 63.2% (*n* = 43 out of 68), rectum 63.1% (*n* = 12 out of 19), small bowel 60.0% (*n* = 3 out of 5), anus 100% (*n* = 2 out of 2), and liver 100% (*n* = 1 out of 1), whereas among MiNENs from the biliary tract (male 47.8%; *n* = 11 out of 23) and the appendix (male 51.0%; *n* = 127 out of 249) the two genders were represented in roughly equal proportions.

The stage of the disease at diagnosis was noted in 77 studies (*n* = 2117) [11,12,13,14,15,16,17,18,19,20,21,22,23,24,25,26,27,28,29,30,31,32,33,34,35,36,37,38,39,40,41,42,43,44,45,46,47,48,49,50,51,52,53,54,55,56,57,58,59,60,61,62,63,64,65,66,68,69,70,73,74,75,79,80,84,85,86,87,88,91,94,95,96,98,99,100,101], and could be classified as follows; localised (Loc), curatively treated, with or without loco-regional nodal involvement, without distant metastases (81.6%; *n* = 1727); advanced (Adv), not suitable for curative treatment, and with or without distant metastases (18.4%; *n* = 390).

The quantitative composition of the primary tumour was described in 36 studies (*n* = 294) [12,15,17,19,21,23,24,25,27,29,30,34,36,39,40,41,45,46,48,49,50,52,53,54,56,57,59,65,73,74,80,82,84,85,86,92]; the two components were present in equal proportion in 27.9% (*n* = 82) of cases, whereas in the remaining 72.1% one of the two histologies was predominant; the neuroendocrine component in 42.2% (*n* = 124), the non-neuroendocrine component in 29.9% (*n* = 88).

Among 69 studies (*n* = 667) reporting the grade of differentiation of the neuroendocrine component [11,12,13,14,15,16,17,18,19,21,22,23,24,25,26,27,28,29,31,32,33,34,35,36,37,38,39,40,41,42,43,44,45,46,47,48,49,50,51,52,54,55,56,57,58,59,60,61,62,68,71,73,74,75,79,80,82,84,85,86,87,89,91,92,93,96,97,100,101], a large proportion of MiNENs (92.5%; *n* = 617) had a grade 3 neuroendocrine component, whereas 4.3% (*n* = 29) and 3.1% (*n* = 21) had a grade 1 and a grade 2 neuroendocrine component, respectively. Among MiNENs with a grade 3 neuroendocrine component, the morphological subtype (large cell or small cell) was reported in 25 studies (*n* = 241) [11,19,21,22,28,29,32,35,38,40,41,46,47,49,52,55,57,58,59,62,79,85,86,89,97], and was large cell in 82.2% (*n* = 198) and small cell in 17.8% (*n* = 43).

The histology of the non-neuroendocrine component was reported in 74 studies (*n* = 606) [11,12,13,15,16,17,18,19,20,21,22,23,24,25,26,27,28,29,30,31,32,33,34,35,36,37,38,39,40,41,42,43,44,45,46,47,48,49,50,51,52,53,54,55,56,57,58,59,60,61,62,64,65,68,72,73,74,75,79,80,81,82,84,85,86,87,89,91,92,93,95,96,97,101], and was consistent with an adenocarcinoma in 92.2% of cases (*n* = 559) (acinar cell carcinoma in 7.6% (*n* = 46)), an adenoma in 4.5% (*n* = 27), a squamous cell carcinoma in 2.5% (*n* = 15), a hepatocellular carcinoma in < 1% (*n* = 1), and a mixture of an adenocarcinoma and a squamous cell carcinoma in < 1% (*n* = 4). The grade of differentiation of the non-neuroendocrine component was specified in 38 studies (*n* = 124), and was well differentiated in 24.2% (*n* = 30), moderately differentiated in 35.5% (*n* = 44), and poorly differentiated in 39.5% (*n* = 49). In one case, the non-neuroendocrine component was described as occupied by a well-differentiated adenocarcinoma and a moderately differentiated squamous cell carcinoma.

Interestingly, in 43 studies (*n* = 61) where more than one diagnostic sample was available [11,13,14,15,16,17,18,19,20,21,22,23,24,25,26,27,28,30,32,33,34,36,37,38,39,41,42,43,44,45,46,47,49,50,53,55,59,60,61,62,64,65,96], the initial diagnosis from the first sample collected (either cytological or histological) was in keeping with MiNEN or suspicion of MiNEN in 36.1% (*n* = 22) of cases, adenocarcinoma in 36.1% (*n* = 22), poorly differentiated neuroendocrine carcinoma in 21.3% (*n* = 13), and well differentiated neuroendocrine tumour in 6.6% (*n* = 4).

Additional data on clinical-pathological characteristics, treatment modalities, and survival outcomes of patients with a diagnosis of MiNEN are presented in Table 1, Table 2 and Table 3 and Appendix A.

Among RSs, 18 (*n* = 571) reported information on treatment modalities [68,69,73,74,75,78,80,84,85,86,88,91,94,96,98,99,100,101] which is illustrated in Figure 2. The great majority of patients received surgery (92.5%; *n* = 528); 66.9% (*n* = 353) in the curative setting, and 13.8% (*n* = 73) in the palliative setting. For the remaining 19.3% (*n* = 102), the disease stage at the time of the surgery remained unknown.

Among RS, 26 (*n* = 2176) reported on survival outcomes of MiNEN [66,68,69,70,72,73,74,75,76,78,80,82,83,84,85,86,87,88,90,94,95,96,98,99,100,101]; in the localised setting, the median recurrence free survival ranged between 8.6 and 75 months, and the median overall survival ranged between 14 and 75 months; in the advanced setting, the progression free survival was 4.6–5.2 months, and the median overall survival was 10–18 months. In studies where both localised and advanced MiNENs were included or the disease stage was not specified, the median overall survival of the whole population ranged between 10.5 and 78 months.

The histology of synchronous or metachronous distant metastases was reported in 14 studies (*N* = 51) [11,17,22,31,50,57,58,62,63,64,85,86,92,96], and was consistent with a single or predominant poorly differentiated neuroendocrine component in 60.8% (*n* = 31) of cases, with a mixture of a neuroendocrine carcinoma and an adenocarcinoma in 33.3% (*n* = 17), and a single or predominant adenocarcinoma component in 5.9% (*n* = 3).

Ten studies investigated the prognosis of MiNENs in comparison with other neoplasms from the same sites of origin (Appendix A); whilst it seems well recognised that patients with an MiNEN diagnosis carry a worse prognosis than patients with well differentiated neuroendocrine tumours [69,70,94,98], it remains controversial whether MiNENs have a better prognosis or not than pure neuroendocrine carcinomas [72,83,85,88,94,96,98,101]. Compared to appendiceal goblet cell carcinoids (more recently defined as goblet cell adenocarcinomas), MiNENs seem to have less favourable survival outcomes [69,70].

### 3.2. The Molecular Landscape of MiNEN and Pathogenetic Hypotheses

Twenty studies (*n* = 381) reported on the genetic/molecular alterations underlying MiNEN [29,30,35,58,59,64,67,71,72,77,79,81,82,85,89,91,92,93,95,97]. In 49.1% of cases where genetic/molecular data was available, the site of origin of MiNEN was the colon–rectum. Most frequent alterations in MiNEN involved well-characterised cancer gene drivers and/or their protein products, such as *TP53* (tumour protein p53), *RB1* (retinoblastoma tumour corepressor 1), *PTEN* (phosphatase and tensin homolog), *APC* (adenomatous polyposis coli)*, PI3KCA* (phosphatidylinositol-4,5-bisphosphate 3-kinase catalytic subunit alpha), *KRAS* (Kirsten rat sarcoma viral oncogene homolog)*, BRAF* (v-raf murine sarcoma viral oncogene homolog B), and MYC (v-myc avian myelocytomatosis viral oncogene homolog) [frequencies of these alterations in individual studies are presented in Table 4]. Activation of the prostaglandin E2 receptor 4 (PTGER4) [95], and microsatellite instability (MSI) have also been proposed as putative driver events of MiNEN.

In the majority of cases where the neuroendocrine and non-neuroendocrine components of MiNEN could be analysed separately, the two components exhibited a core of common alterations, supporting the hypothesis of their common clonal origin, but also alterations exclusively present in one or the other the two components [29,30,58,79,95,97], suggesting that at some point of the tumourigenic process, two distinct morphology entities emerge through the activation of separate genetic programmes. Usually, shared mutations involved well-characterised cancer drivers (e.g., *TP53*, *APC, KRAS, BRAF*) and have higher allele frequencies (compared to alterations which are exclusive of a single component), suggesting their occurrence in the earlier stages of the development of MiNENs [29,58,79,93,97]. In support of this, Yuan et al. performed multiregional next-generation sequencing analysis on samples from spatially separated regions from two patients with oesophageal MiNEN to interrogate intra-tumour heterogeneity and clonal evolution; alterations in *TP53, RB1, PTEN, PI3KCA,* and *KRAS* were identified in all tumour samples/regions from both patients and had higher allele frequencies (compared to alterations not present in all samples/regions). The authors defined these alterations as ‘trunk’, as they were shared by all tumour clones and were likely involved in initiating the tumourigenic process [35]. Compared to the non-neuroendocrine component, the neuroendocrine component usually carried a higher number of aberrations and a higher allele imbalance [30,58,97], which are suggestive of a more aggressive biology. Some authors postulated that the non-neuroendocrine component may give rise to the neuroendocrine component through a trans-differentiation process and the acquisition of a more aggressive phenotype [30,58,81]. c-Myc and SMARC4A have been indicated as potential mediators of this trans-differentiation process [58,81]. In some cases, the two components exhibited fairly distinct patterns of genetic or chromosomal alterations [93,95,97], raising the possibility of a polyclonal origin for at least a subtype of MiNENs. Interestingly, in the study by La Rosa et al. including only MiNENs composed of an adenoma and a well-differentiated neuroendocrine component, no *KRAS, BRAF,* or *PI3KCA* mutation or MSI was found in either components of all four samples analysed [82].

With regards to the comparison between MiNENs and their pure counterparts at a genetic/molecular level, Sinha et al. reported that colonic MiNENs (*n* = 14) and pure colonic adenocarcinomas (*n* = 269) shared a largely similar copy number aberration (CNA) profile, whereas pure colonic neuroendocrine carcinomas (*n* = 5) displayed distinct structural chromosomal alterations, suggesting that MiNENs may have a closer developmental relationship to adenocarcinomas than to neuroendocrine carcinomas [95]. Likewise, Jesinghaus et al. reported that colorectal MiNENs (*n* = 19) exhibited a genetic/molecular profile broadly similar to that of pure colorectal adenocarcinomas, but lack alterations commonly related to pure neuroendocrine carcinomas of various origins [79].

## 4. Discussion

This systematic review comprises the largest collection of studies on MiNEN available in the current literature. Overall, evidence available is of poor quality; the studies included are CRs or RSs (neither published nor ongoing (https://clinicaltrials.gov/) prospective trials specifically recruiting patients with a diagnosis of MiNEN were identified), and are extremely heterogeneous in terms of site of origin of the primary tumour, disease stage, geographical area of patients included, and type of information provided (clinical-pathological data, treatment modalities, survival outcomes, genomic/molecular findings). Furthermore, the 2010 WHO classification was ambiguous as to whether adenocarcinomas ex-goblet cell carcinoids (goblet cell carcinoids Tang B and C) could be regarded, or not, as MANEC [4], generating an additional source of inconsistency within the published literature. Therefore, it was not possible to completely rule out the inclusion of these entities in the RSs of the present review, especially in the two largest reporting on appendiceal MiNEN by Brathwaite et al. (*n* = 249) and by Mehrvarz Sarshekeh et al. (*n* = 1173), and this may have introduced a further confounding element.

Acknowledging these limitations, this review suggests that the biological behaviour of MiNENs is mostly driven by the neuroendocrine component, which is poorly differentiated in approximately 90% of cases, and often occupies the distant metastatic sites. This was also corroborated by molecular findings showing that the neuroendocrine component exhibits more genetic and chromosomal alterations.

Regarding treatment modalities, surgery was the treatment of choice for nearly all potentially curable cases, and was also offered to approximately a quarter to a third of patients with advanced disease. In the latter setting, surgery was pursued for symptom relief or with initial curative intent in patients subsequently found to have advanced disease. However, in most cases, the reasons supporting the choice for surgery in the palliative setting remains unknown. Adjuvant, neoadjuvant, or perioperative therapies were offered to a third of patients receiving curative surgery. The choice of perioperative chemotherapy regimen was most often based on the clinical practice guidelines for early stage adenocarcinomas from the same sites of origin; in fact, the use of (neo) adjuvant chemotherapy protocols to prevent/delay the relapse of the neuroendocrine component is not supported by randomised evidence from the perioperative setting of pure grade 2 or 3 neuroendocrine neoplasms [8]. Palliative chemotherapy was delivered to between a half and two thirds of patients with advanced disease, as upfront treatment, or after palliative surgery. Regimens of palliative systemic treatments were chosen according to the standard of care for either pure adenocarcinomas or neuroendocrine carcinomas from the same site of origin in roughly equal proportion. Noticeably, among the clinical practice guidelines from international oncology societies screened, only the ENETS guidelines provide indications on the treatment for patients with a MiNEN diagnosis, and suggest treatment algorithms based on those used in pure neuroendocrine carcinomas [8], probably because of the aggressiveness of the neuroendocrine component in the majority of cases.

Survival outcomes in the localised setting were largely variable across RSs, ranging from a few months to several years, likely due to differences in patient selection criteria and follow-up time. Often, median survival times were not reached in the localised setting due to the lack of long-term follow-up data. This is also reflected in the initial paper selection of the review, where a large proportion of publications was discarded because information on the patient/disease status was completely missing or limited to a short period (<6 months) after initial diagnosis; longer follow-up data should be obtained when reporting on patients with MiNENs to allow for a more reliable estimation of the prognosis of this disease, especially when still potentially curable. In contrast, in the advanced setting, survival outcomes were more consistent across RSs and with those estimated for CRs (median progression free survival of 5–6 months and a median overall survival of 12–18 months), and very close to those of advanced pure neuroendocrine carcinomas [8]. This further supports the putative similarity in biological behaviour between MiNENs and pure neuroendocrine carcinomas in the advanced setting.

The limitation of biopsy samples in diagnosing MiNENs is a critical issue. Biopsies may not accurately distinguish MiNENs from their pure counterparts, especially because this discrimination depends on a quantitative threshold. In fact, in the present review, the initial biopsy was able to identify the presence of a mixed histology in only a third of cases. This can be due to either the paucity of tumour tissue in the biopsy sample, not representative of both histologies, or because the biopsy is performed on metastatic site, most commonly occupied by only one of the two components. As a further demonstration, in the current review, only around 1 out of 5 patients presented with advanced stage at diagnosis; a much higher proportion of advanced cases would be expected for a highly aggressive disease, and this may be due to the limited ability of biopsy to diagnose advanced MiNENs when not amenable to surgical resection. There is also controversy surrounding the validity of the 30% threshold as discriminatory criterion between MANECs/MINENs and their pure counterparts. Whether the presence of elements with neuroendocrine differentiation within predominantly exocrine neoplasms, or vice versa, affects the outcome of patients and informs clinical decision making, and whether specific cut-offs in the proportions of each component account for different prognoses and responses to treatment, represent unanswered questions. Some studies have demonstrated that alternative thresholds (e.g., < versus > 10% or 20%) identify adenocarcinomas associated with a minor neuroendocrine component as having a significantly better prognosis than neoplasms with a proportion of neuroendocrine component above those thresholds [86,88]. La Rosa et al. proposed a solution to partially overcome this issue [2]; they suggested that if there is a suspicion of a mixed neuroendocrine/non-neuroendocrine neoplasm within a tumour sample, further confirmation should be pursued through immunohistochemical analysis.

Studies reporting on molecular/genetic data of GEP MiNENs have identified well-characterised carcinogenetic hallmarks of more common GEP malignancies as potential drivers of this disease, such as alterations affecting *TP53, KRAS, BRAF, APC, PI3KCA*, and MSI [102], corroborating what was previously reported by Girardi D.M. et al. [10]; these alterations are usually shared between the two components and likely present in founding clones. Although displaying a biological behaviour more similar to that of pure neuroendocrine carcinomas, the molecular/genetic landscape of GEP MiNENs seems to be closer to that of pure adenocarcinomas. This supports the hypothesis according to which the two components of MiNEN may arise from common glandular precursor through similar sequences of aberrant events to those driving pure GEP adenocarcinomas [79,97]. At a later stage of the tumorigenesis, the two components separate and evolve independently, with the neuroendocrine one accumulating more aberrations and acquiring a more lethal phenotype. Current available data does not allow clarification as to whether the neuroendocrine component arises through trans-differentiation of the non-neuroendocrine one or the two components develop independently. Either way, these findings open new avenues for the exploration of targeted treatments and immunotherapies with already proven activity in the treatment of GEP adenocarcinomas.

In conclusion, the MiNEN is likely an underestimated disease, due to the controversies relating to its definition, the limited diagnostic ability of biopsies, and the lack of awareness of this diagnosis within the scientific community (suggested by the absence of clinical trials enrolling patients with this diagnosis, and the minimal referencing by major international oncology societies). To increase the likelihood of diagnosing MiNEN, core biopsies should be sought when a surgical sample is not available, and analysed by pathologists with expertise in neuroendocrine neoplasms.

Because of the low quality of the evidence collected, it is very difficult to formulate recommendations on the best management of patients with an MiNEN diagnosis. Therefore, newly diagnosed patients with MiNENs should be discussed within multidisciplinary meetings and the treatment strategy should be planned on the basis of the most aggressive and/or predominant component in the diagnostic sample. Following the standard practice for their pure counterparts is entirely appropriate given that randomised studies are unlikely to be feasible in this patient group. Furthermore, since only one of the two components is present in most distant metastatic sites, the collection of a second tumour sample is advisable to optimise the management and guide the choice of systemic treatment in the following scenarios; (1) in the presence of synchronous distant metastases when the original sample is from the primary tumour, (2) on metastatic recurrence of a previously resected MiNEN, and (3) on development of new/rapidly growing metastatic lesions while on treatment, in the setting of otherwise stable disease. The advent of liquid biopsies may aid in delivering more customised treatments for these diseases. Genomic profiling of tumour or blood samples of patients diagnosed with an MiNEN should be encouraged, with a view to widening the knowledge of the biology of this disease and possibly offering those patients participation in prospective early phase or basket type/umbrella clinical trials.

## Figures and Tables

**Figure 1 jcm-09-00273-f001:**
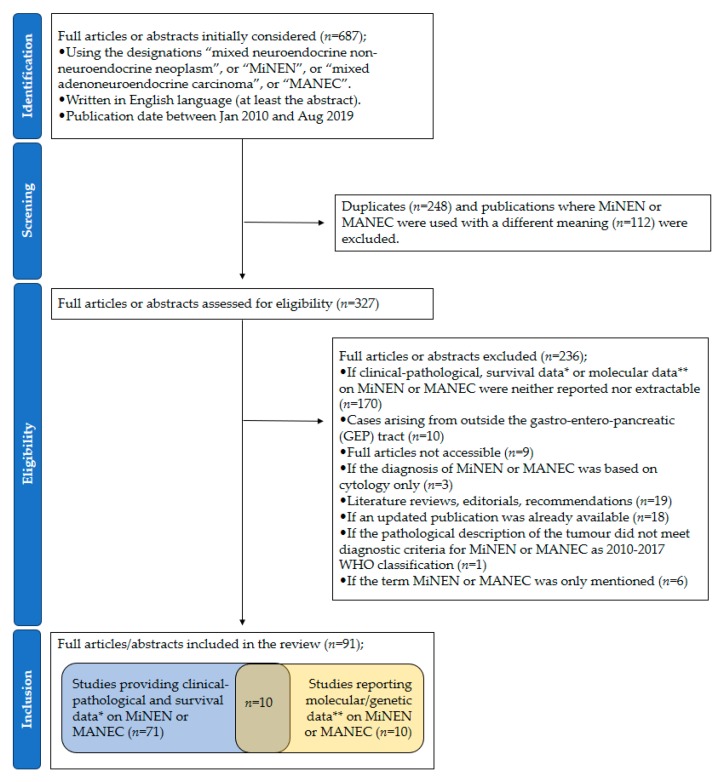
PRISMA flow diagram of study selection. PRISMA = Preferred Reporting Items for Systematic Reviews and Meta-Analyses; *n* = number of studies; MiNEN = mixed neuroendocrine non-neuroendocrine neoplasm; MANEC = mixed adenoneuroendocrine carcinoma; * follow-up time ≥ 6 months for patients who were alive at the time of publication; ** Immunohistochemical data were not included, except when used to assess DNA mismatch repair protein status.

**Figure 2 jcm-09-00273-f002:**
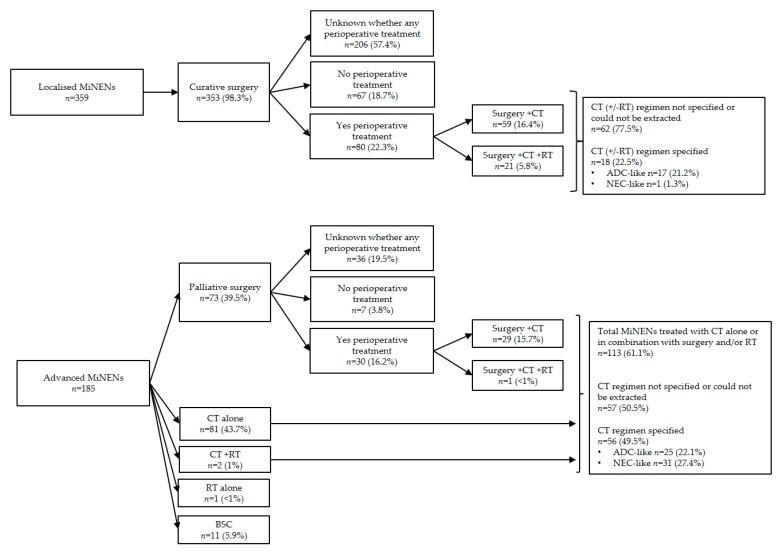
Treatment modalities of MiNEN in retrospective studies. MiNEN = mixed neuroendocrine non-neuroendocrine neoplasm; *n* = number of patients; CT = chemotherapy; RT = radiotherapy; BSC = best supportive care; peri operative = pre surgical and/or post-surgical; ADC-like = in keeping with standard of care for pure adenocarcinomas from the same sites of origin; NEC-like = in keeping with standard of care for pure neuroendocrine carcinomas.

**Table 1 jcm-09-00273-t001:** Clinical-pathological characteristics and survival outcomes of patients with a diagnosis of mixed neuroendocrine non-neuroendocrine neoplasm in case reports.

Characteristics	All Patients (*n* = 61)
**Gender**	
Male	47 (77.1%)
Female	14 (22.9%)
**Age at diagnosis (median)**	64 years
**Primary tumour site**	
Stomach	23 (37.3%)
Oesophagus/OGJ	5 (8.2%)
Pancreas	2 (3.3%)
Biliary tract	15 (24.6%)
Colon	11 (18.0%)
Rectum	3 (4.9%)
Small bowel	1 (1.6%)
Liver	1 (1.6%)
**Ki-67 NE component (median)**	70% (available for 41 patients)
**Disease stage at diagnosis**	
Localised	48 (78.7%)
Advanced	13 * (21.3%)
**Survival outcomes**	
*n* (%) of patients with survival data	59 (96.7%)
*n* (%) of recurrence events	18 (37.5%)
*n* (%) of death events	21 (35.6%)
Follow-up time (median)	14.5 months
Overall Survival (median)	35 months (95%CI could not be estimated)
**Sites of recurrence (localised stage cases)**	11 †
Liver	7 (63.6%)
Retroperitoneal lymph nodes	2 (18.1%)
Peritoneum	1 (9.0%)
Lung	1 (9.0%)
Supraclavicular lymph node	1 (9.0%)
Scalp	1 (9.0%)
**Sites of progression (advanced stage cases)**	7 ‡
Liver	5 (71.4%)
Local recurrence after palliative surgery	1 (14.3%)
Not reported	1 (14.3%)

*n* = number of patients; OGJ = oesophagogastric junction; NE = neuroendocrine; 95%CI = 95% confidence interval; * 31 patients included in survival analysis for the advanced stage subgroup (13 patients with advanced disease at diagnosis plus 18 patients who recurred after initial curative treatment for localised disease); † sites of disease recurrence were reported for 11 out of 18 patients with localised diseases who developed recurrence; ‡ information on disease status at the last follow-up was available for 9 out of 13 patients with advanced disease at diagnosis, and 7 out of these 9 patients had documented progression.

**Table 2 jcm-09-00273-t002:** Treatment modalities and survival outcomes of patients with a diagnosis of mixed neuroendocrine non-neuroendocrine neoplasm in case reports according to disease stage.

Localised (*n* = 48)		Advanced (*n* = 13)	
**Primary Tumour Site**		**Primary Tumour Site**	
Upper gastro-intestinal tract	25 (52.1%)	Upper gastro-intestinal tract	3 (23.1%)
Lower gastro-intestinal tract	8 (16.7%)	Lower gastro-intestinal tract	7 (53.8%)
Hepato-pancreato-biliary tract	15 (31.2%)	Hepato-pancreato-biliary tract	3 (23.1%)
**Curative treatment**		**Palliative treatment (*n* = 31 *)**	
Surgery alone	29 (60.4%)	Surgery alone	2 (6.5%)
Surgery + CT	16 (33.3%)	Surgery + CT	6 (19.4%)
Surgery + CT + RT	3 (6.3%)	CT + RT	2 (6.5%)
		CT alone	9 (29.0%)
		RT alone	2 (6.5%)
		Best supportive care	4 (12.9%)
		Unknown	6 (19.4%)

Curative surgery	48 (100%)	Palliative surgery	8 (25.8%)
Perioperative CT or CT/RT	19 (39.6%)	Palliative CT	17 (54.8%)
**CT regimen (+/−RT)**		**CT regimen (+/−Surgery +/−RT)**	
Platinum/Etoposide	3 (15.8%)	Platinum/Etoposide	6 (35.3%)
Platinum/Irinotecan	1 (5.2%)	Platinum/Irinotecan	1 (5.9%)
Fluoropyrimidine/Platinum/Irinotecan	1 (5.2%)	Fluoropyrimidine/Platinum	1 (5.9%)
Fluoropyrimidine/Platinum/Etoposide	1 (5.2%)	Fluoropyrimidine/Oxaliplatin (+/−mAb)	3 (17.6%)
Fluoropyrimidine/Oxaliplatin	8 (42.1%)	Fluoropyrimidine/Irinotecan (+/−mAb)	2 (11.8%)
Fluoropyrimidine alone	2 (10.5%)	Fluoropyrimidine alone	1 (5.9%)
Gemcitabine/Oxaliplatin	2 (10.5%)	Gemcitabine	1 (5.9%)
Regimen not specified	1 (5.2%)	Regimen not specified	2 (11.8%)
Non-NE-like regimens	12 (66.7%)	Non-NE-like regimens	8 (53.3%)
NEC-like regimens	4 (22.2%)	NEC-like regimens	7 (46.7%)
Both NEC-like and non-NE-like regimens	2 (11.1%)	Both NEC-like and non-NE-like regimens	0
**Median RFS (95%CI)** (could be estimated for 48 patients)	36 m (95%CI; 5.8–66.2)	**Median PFS (95%CI)** (could be estimated for 17 patients)	5 m (95%CI; 3.6–6.4)
**Median OS (95%CI)** (could be estimated for 48 patients)	N.R.	**Median OS (95%CI)** (could be estimated for 20 patients)	12m (95%CI; 4.4–19.6)

*n* = number of patients; CT = chemotherapy; RT = radiotherapy; Platinum = cisplatin or carboplatin; mAb = monoclonal antibody; NEC-like regimens = regimens recommended for pure neuroendocrine carcinomas; non-NE-like regimens = regimens recommended for pure non-neuroendocrine malignancies (most commonly adenocarcinomas or squamous cell carcinomas) from the same site of origin; RFS = recurrence free survival; PFS = progression free survival; OS = overall survival; m = months; 95%CI = 95% confidence interval. * 13 patients with advanced disease at diagnosis plus 18 patients who relapsed after initial curative treatment.

**Table 3 jcm-09-00273-t003:** Treatment modalities and survival outcomes of patients with a diagnosis of mixed neuroendocrine non-neuroendocrine neoplasm in retrospective studies.

Reference	Primary Tumour Site	*n* pts	Age at Diagnosis ‡	*n* (%) Localised	*n* (%) Advanced	Ki-67 NE Component ‡	Treatment for Localised Disease	Treatment for Advanced Disease	Median RFS	Median OS (Localised)	Median PFS	Median OS (Advanced)	Median OS (Whole Population)
Shen C., 2016	Stomach	20	62.2 years	14 (70%)	6 (30%)	n.a.	Surgery alone (13; 65%) Surgery + CT (7; 35%) Platinum/Etop or Platinum-irinotecan		n.a.	n.a.	n.a.	n.a.	10.5 m
Lim S.M., 2016	Stomach	17	n.a.	n.a.	n.a.	n.a.	n.a.	n.a.	n.a.	n.a.	n.a.	n.a.	36.4 m
Nie L., 2016	Stomach	14	60.5 years	13 (92.9%)	1 (7.1%)	n.a.	Surgery * (13; 100%)	Surgery * (1; 100%)	N.R. #	N.R. #	not applicable	not applicable	N.R.#
Park J.Y., 2014	Stomach	10	65.5 years	10 (100%)	n.a.	n.a.	Surgery alone (3; 30%) Surgery + CT (7; 70%) Fluorop alone: 2 Fluorop/other **: 1 Fluorop/platinum: 4	n.a.	~75 m @	~75 m @	n.a.	n.a.	~75 m @
Zhang P., 2018	Oesophagus/OGJ	96	62.1 years	82 (85.4%)	14 (14.6%)	66.7%	Surgery * (87; 90.6%)		n.a.	n.a.	n.a.	n.a.	73.3 m
van der Veen A., 2018	Stomach (8) Oesophagus/OGJ (9)	17	64.3 years	17 (100%)	n.a.	n.a.	Surgery alone (8; 47%) Surgery + CT (5; 29.4%) Anthracycline/platinum: 5 Surgery + CT + RT (4; 23.5%) Platinum/etoposide: 1 Platinum/taxane: 2 Other**: 1	n.a.	n.a.	Stomach: ~14 m @ Oesophagus/OGJ: ~44 m @	n.a.	n.a.	Stomach: ~14 m @ Oesophagus/OGJ: ~44 m @
Basturk O., 2014	Pancreas	28	n.a.	n.a.	n.a.	n.a.	n.a.	n.a.	n.a.	n.a.	n.a.	n.a.	44 m
Schimmack S., 2017	Pancreas	11	n.a.	8 (72.7%)	3 (27.3%)	n.a.	n.a.	n.a.	n.a.	n.a.	n.a.	n.a.	60 m
Yang M., 2016	Pancreas	6	47 years	3 (50%)	3 (50%)	n.a.	Surgery * (3; 100%)	Surgery * (3; 100%)	n.a.	n.a.	n.a.	n.a.	15 m
Pop G., 2016	Pancreas	5	n.a.	n.a.	5 (100%)	n.a.	n.a.	Surgery * (5; 100%)	n.a.	n.a.	n.a.	10 m	10 m
Zheng Z., 2019	Biliary tract	6	62 years	6 (100%)	n.a.	70%	Surgery alone (6; 100%)	n.a.	9.5 m	23 m	n.a.	n.a.	23 m
La Rosa S., 2018	Pancreas	4	n.a.	n.a.	n.a.	n.a.	n.a.	n.a.	n.a.	n.a.	n.a.	n.a.	n.a.
Olevian D., 2015	Colon	26	n.a.	n.a.	n.a.	n.a.	n.a.	n.a.	n.a.	n.a.	n.a.	n.a.	n.a.
Jesinghaus M., 2017	Colon	19	64.3 years	n.a.	n.a.	n.a.	n.a.	n.a.	n.a.	n.a.	n.a.	n.a.	n.a.
Kolasinska-Cwikla AD., 2016	Colon	15	n.a.	n.a.	n.a.	26.9%	Surgery * (15; 100%)		n.a.	n.a.	n.a.	n.a.	26 m
Sinha N., 2018	Colon	14	73.5 years	5 (35.7%)	9 (64.3%)	n.a.	n.a.	n.a.	n.a.	n.a.	n.a.	11 m †	11 m †
Lee S.M., 2016	Colon	8	n.a.	n.a.	n.a.	n.a.	n.a.	n.a.	n.a.	n.a.	n.a.	n.a.	n.a.
Bongiovanni M., 2017	Colon	6	n.a.	6 (100%)	n.a.	<2%	Surgery * (6; 100%)	n.a.	n.a.	n.a.	n.a.	n.a.	n.a.
Woischke C., 2017	Colon (10) Rectum (5)	15	72 years	n.a.	n.a.	71%	n.a.	n.a.	n.a.	n.a.	n.a.	n.a.	n.a.
Komatsubara T., 2016	Colon (5) Rectum (1)	6	69 years	4 (66.7%)	2 (33.3%)	50%	Surgery alone (1; 25%) Surgery + CT (3; 75%) Fluorop alone: 3	Surgery + CT (2; 100%) Fluorop alone: 1 Fluorop/Irinotecan: 1	n.a.	n.a.	n.a.	n.a.	53 m *
Brathwaite S., 2016	Appendix	249	58 years	176 (70.7%)	73 (29.3%)	n.a.	n.a.	n.a.	n.a.	n.a.	n.a.	18 m	78 m
Mehrvarz Sarshekeh A., 2016	Appendix	1173	n.a.	1034 (88.2%)	139 (11.8%)	n.a.	n.a.	n.a.	n.a.	Stage I (52 m), Stage II (43 m), Stage III (28 m)	n.a.	17 m	17 m (stage IV)–52 m (stage I)
Milione M., 2018	Stomach (32) Oesophagus/OGJ (12) Pancreas (14) Biliary tract (10) Colon (74) Rectum (18)	160	n.a.	143 (89.4%)	17 (10.6%)	≥55%: 82.5% <55%: 17.5%	Surgery * (143; 100%)	Surgery * (17; 100%)	n.a.	n.a.	n.a.	n.a.	13.2 m
Yin X.N., 2018	Stomach (20) Rectum (6) Small bowel (4) Appendix (1)	31	61 years	27	4	n.a.	Surgery * (27; 100%)	Surgery * (4; 100%)	n.a.	n.a.	n.a.	n.a.	13 m @
Apostolidis L., 2018	Stomach (6) Oesophagus/OGJ (11) Pancreas (14) Biliary tract (4) Colon-rectum (44) Small bowel (6) Anus (1) Non-GEP (9)	96	59 years	61 (63.5%)	35 (36.5%) (68 for survival analysis)	78%	Surgery alone (23; 37.7%) Surgery + CT ** (25; 40.9%) Surgery + CT + RT ** (9; 14.8%) Unknown treatment (4; 6.6%)	CT alone (54; 79.4%) NEC-like: 31 ADC-like: 23 Unknown treatment (14; 20.6%)	8.6 m (surgery alone) − 12.9 m (surgery + periop)	18.9 m (surgery alone) − 75m (surgery + periop)	5.2 m	17.4 m	44.5 m
Düzköylü Y., 2018	Stomach (5) Pancreas (1) Biliary tract (2) Colon (1) Rectum (1)	10	67.5 years	9 (90%)	1 (10%)	55.5%	Surgery alone (5; 55.67%) Surgery + CT ** (4; 44.4%)	Surgery + CT ** (1; 100%)	n.a.	N.R.#	n.a.	n.a.	N.R.#
Frizziero M., 2017	Stomach (3) Oesophagus/OGJ (10) Pancreas (3) Biliary tract (2) Colon-rectum (31) Small bowel (3) Unknown (1)	53	62 years	28 (52.8%)	25 (47.2%) (41 for survival analysis)	70%	Surgery alone (12; 42.9%) Surgery + CT ** (7; 25%) Surgery + CT +RT ** (7; 25%) CT + RT (1; 3.6%) Unknown treatment (1; 3.6%)	CT alone (27; 65.9%) Platinum-based: 20 Irinotecan-based: 3 Gemcitabine: 1 Others **: 3 CT + RT (1; 2.4%) RT alone (1; 2.4%) Best Supportive care (11; 26.8%) Unknown treatment (1; 2.4%)	19.4 m	21 m	4.6 m	13.6 m	18.6 m
La Rosa S., 2018	Stomach (2) Colon (4) Rectum (5) Small bowel (3)	14	57.5 years	n.a.	n.a.	1%	n.a.	n.a.	n.a.	n.a.	n.a.	n.a.	N.R.#
Scardoni M., 2014	Stomach (2) Pancreas (2) Rectum (1) Small bowel (1)	6	68 years	n.a.	n.a.	65%	n.a.	n.a.	n.a.	n.a.	n.a.	n.a.	n.a.
Dulskas A., 2019	Colon (4) Rectum (3) Anus (2)	9	61 years	3 (33.3%)	6 (66.7%)	65%	Surgery alone (2; 66.7%) Surgery + CT + RT (1; 33.3%)	Surgery alone (1; 16.7%) Surgery + CT (3; 60%) Surgery + CT + RT (1; 20%) CT + RT (1: 60%)	n.a.	n.a.	n.a.	n.a.	N.R.#
							Fluorop/Oxaliplatin: 3 Platinum/Etoposide: 2 Fluorop alone: 1						
Spada F., 2019	Colon-rectum (32) GEP n.o.s. (19)	51	n.a.	n.a.	n.a.	n.a.	n.a.	n.a.	n.a.	n.a.	n.a.	n.a.	14.4 m
Brathwaite S., 2016	Colon (4) Appendix (40) Small bowel (1) Unknown (1)	46	54 years	15 (32.6%)	31 (67.4%)	n.a.	Surgery alone (7; 46.7%) Surgery + CT ** (8; 53.3%)	Surgery alone (6; 19.4%) Surgery + CT ** (23; 74.2%) Unknown treatment (2; 6.4%)	n.a.	n.a.	n.a.	n.a.	49.2 m
Bu S., 2017	GEP n.o.s.	19	n.a.	n.a.	n.a.	n.a.	n.a.	n.a.	n.a.	n.a.	n.a.	n.a.	25 m @
Melchior L.C., 2019	GEP n.o.s.	43	n.a.	n.a.	n.a.	n.a.	n.a.	n.a.	n.a.	n.a.	n.a.	n.a.	n.a.
Sahnane N., 2015	GEP n.o.s.	36	n.a.	n.a.	n.a.	n.a.	n.a.	n.a.	n.a.	n.a.	n.a.	n.a.	n.a.
Yang H.-M., 2015	GEP n.o.s.	27	n.a.	n.a.	n.a.	n.a.	n.a.	n.a.	n.a.	n.a.	n.a.	n.a.	n.a.

*n* = number; Fluorop = fluoropyrimidine; CT = chemotherapy; RT = radiotherapy; periop = perioperative CT and/or RT; OGJ = oesophagogastric junction; GEP = gastroenteropancreatic tract; n.o.s. = not otherwise specified (primary tumour arising within the gastroenteropancreatic tract but the exact organ of origin was not specified or could not be extracted); NE = neuroendocrine; NEC = neuroendocrine carcinoma; ADC = adenocarcinoma; n.a. = information not available or could not be extracted; N.R.# = not reached (further information provided in Appendix A); RFS = recurrence free survival; PFS = progression free survival; OS = overall survival; m = months; ‡ mean or median; * unknown whether any perioperative treatment; ** regimen not specified; † survival estimated by applying Kaplan-Meier analysis to data provided in the publication; @ survival estimation extracted from Kaplan–Meier curves.

**Table 4 jcm-09-00273-t004:** Molecular data on patients with mixed neuroendocrine non-neuroendocrine neoplasms.

Reference	Primary Tumour Site	n pts	Method(s)	Molecular Findings
Fujita Y., 2019	Stomach	1	PCR, DNA methylation analysis	*TP53* mutation: absent in either components. Low DNA methylation status in either components. Allele imbalance (AI) on chromosomes 5q, 8p, 11q and 22q in NEC, AI on chromosome 11q in ADC.
Farooq F., 2018	Stomach	1	Targeted NGS (255 cancer-related genes—Foundation Medicine)	Tumour with trilineage differentiation (NEC, ADC, SCC) *KRAS, NF1, CDKN2A/B, TP53* mutations: present in all 3 components (same mutation). MSI status: negative in all 3 components. Low TMB in all 3 components. *CDK6, PIK3CG, TOP2A* amplification: present only in NEC. Loss of *PTEN* exons 1–2: present in ADC and SCC (not in NEC). *NOTCH1* mutation: present only in ADC *TERT* amplification: present only in SCC.
Yuan W., 2017	Oesophagus/OGJ	2	Whole exome sequencing, whole genome single nucleotide polymorphism	Multiregional next-generation sequencing *TP53* and *NOTCH1* mutation: present in 2/2 (100%)—all regions analysed. *RB1* deletion or LOH: present in 2/2 (100%)—all regions analysed. *PI3KCA, PTEN, KRAS, SOX2, DVL3, TP63* amplification: present in 2/2 (100%)—all regions analysed.
Basturk O., 2014	Pancreas	6	Not specified	*KRAS* mutation: present in 0/6 (0%).
La Rosa S., 2018	Pancreas	4	Fluorescent in situ hybridisation (FISH)	*MYC* amplification and/or chromosome 8 polysomy: present in all 4 cases.
Vanacker L., 2014	Colon	1	Whole exome sequencing, IHC for MMR proteins	*KRAS, APC, BCL9, FOXP1* mutations: present in both components. *SMARC4A* mutation: present only in NEC. MSI status: negative.
Ito H., 2014	Colon	1	Not specified	*KRAS* mutation: absent (analysed in ADC).
Olevian D., 2015	Colon	26	Not specified	*KRAS* mutation: present in 4 (15.4%). *BRAF* mutation: present in 17 (65.4%).
Jesinghaus M., 2017	Colon	19	Targeted NGS (panel including 196 amplicons covering 32 genes)	*TP53* mutation: present in 9 (47.4%). *KRAS* mutation: present in 4 (21.0%). *BRAF* mutation: present in 7 (36.8%). *APC* mutation: present in 3 (15.8%).*RB1* mutation: present in 1 (5.3%). *PTEN* mutation: present in 2 (10.5%). *ATM* mutation: present in 3 (15.8%). *FBXW7* mutation: present in 3 (15.8%). *SOX9* mutation: present in 2 (10.5%). *MYC* amplification: present in 1 (5.3%). MSI status: positive in 2 (10.5%).
Sinha N., 2018	Colon	14	Genome-wide copy number aberration analysis and FISH	*BRAF* mutation: present in 8 (57.1%). *PTGER4* amplification: present in 1 (7.1%). *MYC* amplification: present in 1 (7.1%). MSI status: positive in 1 (7.1%). CN gains of chr.: 5p; 10/14 (71.4%), 7; 11/14 (78.6%), 8q; 12/14 (85.7%), 13q; 9/14 (64.3%), 20q; 11/14 (78.6%). CN losses of chr.: 3p; 5/14 (35.7%); 4p; 7/14 (50%), 8p; 6/14 (42.9%), 18q; 7/14 (50%).
Lee S.M., 2016	Colon	8	Targeted NGS panel analysing substitutions and small indels in 46/50/409 cancer-related genes	*TP53* mutation: present in 3 (37.5%). *KRAS* mutation: present in 6 (75%). *BRAF* mutation: present in 1 (12.5%). *APC* mutation: present in 3 (37.5%). *RB1* mutation: present in 1 (12.5%). *PTEN* mutation: present in 1 (12.5%). *PI3KCA* mutation: present in 1 (12.5%). *GNAS* mutation: present in 1 (12.5%). *SMO* mutation: present in 1 (12.5%). *FBXW7, CDKN2A, ERBB2, FGFR3, PTPN11* mutation: present in 0 (0%).
Bongiovanni M., 2017	Colon	6	Direct sequencing (not specified)	*KRAS, BRAF, PI3KCA* mutation: present in 0/6 (0%)—absent in either component. MSI status: positive in 0/6 (0%)—absent in either component.
Woischke C., 2017	Colon (10)Rectum (5)	15	PCR, targeted NGS (50 gene panel) and whole exome sequencing	*KRAS* mutation assessed by PCR (in 15 patients): present in 9 (60%). Genes assessed by an NGS panel (in 10 patients): *TP53* mutation: present in 10 (100%) (same mutation in both components: 6/10, distinct mutations in the two components: 2/10, exclusively in NEC: 1/10, exclusively in ADC: in 1/10). *KRAS* mutation: present in 9 (90%) (same mutation in both components: 8/9, exclusively in NEC: 1/9). *BRAF* mutation: present in 2 (20%) (same mutation in both components: 1/2, distinct mutations in the two components: 1/2). *APC* mutation: present in 8 (80%) (same mutation in both components: 7/8, exclusively in NEC: 1/8). *RB1* mutation: present in 3 (30%) (same mutation in both components: 1/3, distinct mutations in the two components: 1/3, exclusively in NEC: 1/3). *PI3KCA* mutation: present in 5 (50%) (exclusively in ADC: 4/5, exclusively in NEC: 1/5). *MET* mutation: present in 4 (40%) (same mutation in both components: 1/4, exclusively in NEC: 2/4, exclusively in ADC: 1/4). *NOTCH1* mutation: present in 3 (30%) (same mutation in both components: 1/3, exclusively in NEC: 2/3). *RET* mutation: present in 2 (20%) (same mutation in both components: 1/2, exclusively present in NEC: 1/2).
Quaas A., 2018	Small Bowel	1	Targeted panel including 14 genes and 14 microsatellite loci	Germline *BRCA-1* mutation: present. MSI status: absent. *TP53* mutation: present. *KRAS, NRAS, HRAS, BRAF, DDR2, ERBB2, KEAP1, NFE2L2, PIK3CA, PTEN, RHO, BRCA2* mutations: absent.
Milione M., 2018	Stomach (32)Oesophagus/OGJ (12)Pancreas (14)Biliary tract (10)Colon (74)Rectum (18)	160	PCR, targeted NGS panel	*TP53* mutation (assessed in 71 patients): present in 17 (23.9%) (assessed in the whole tumour). *KRAS* mutation (assessed in 71 patients): present in 12 (16.9%) (assessed in the whole tumour). *BRAF* mutation (assessed in 71 patients): present in 4 (5.6%) (assessed in the whole tumour). MSI status (assessed in 160 patients): positive in 8 (5%) (in both components).
La Rosa S., 2018	Colon (1)Rectum (3)	4	Direct sequencing (not specified)	*KRAS* mutation: present in 0% (in both components). *TP53* mutation: present in 0% (in both components). *PI3KCA* mutation: present in 0% (in both components). MSI status: positive in 0% (in both components).
Scardoni M., 2014	Stomach (2)Pancreas (2)Rectum (1)Small bowel (1)	6	Targeted NGS (54 gene panel)	*TP53* mutation: present in 6 (100%) (5/6 in both components, same mutation; 1/6 only in ADC). *KRAS* mutation: present in 1 (16.7%) (in both components, same mutation). *RB1* mutation: present in 1 (16.7%) (in both components, same mutation). *ERBB4, ATM, JAK3, KDR* mutations: present in 1/6 (16.7%) (only in NEC). *CTNNB1* mutation: present in 1/6 (16.7%) only in ADC *ATRX, DAXX, MEN1, TSC2* mutations: present in 0/6 (0%).
Melchior L.C., 2019	GEP n.o.s	43	Targeted NGS (50 gene panel)	*TP53* mutation: present in 28 (65.1%). *KRAS* mutation: present in 7 (16.3%). *BRAF* mutation: present in 6 (13.9%).
Sahnane N., 2015	GEP n.o.s	36	PCR, DNA methylation analysis of 34 gene promoters and MMR genes	*KRAS* mutation (assessed in 88 MiNEN and NEC): present in 15 (17%). *BRAF* mutation (assessed in 88 MiNEN and NEC): present in 6 (6.8%) (6 colorectal). Methylation status (assessed in 89 MiNEN and NEC): high levels (>8 methylated genes) in 28 (31.5%). MSI status (assessed in 36 MINEN): positive in 4 (11.1%) (2 stomach and 2 colorectal).
Yang H.-M., 2015	GEP n.o.s	27	Direct sequencing (not specified)	*TP53* mutation: present in 19 (70.4%) (shared by both components in 13/19, only present in NEC in 6/19). *KRAS* mutation: present in 10 (37%) (in both components).

*n* = number; pts = patients; NEC = neuroendocrine carcinoma; ADC = adenocarcinoma; SCC = squamous cell carcinoma; GEP = gastro-enteropancreatic tract; n.o.s. = non-otherwise specified; OGJ = oesophagogastric junction; MSI = microsatellite instability; MMR = mismatch repair; NGS = next-generation sequencing; PCR = polymerase chain reaction; IHC = immunohistochemistry; CN = copy number; chr. = chromosome. MSI status is defined as positive if MSI is detected by PCR in at least two of the microsatellite loci analysed, or if at least one of the MMR proteins (Mlh1, Msh2, Msh6 or Pms2) is not expressed or abnormally expressed on IHC.

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
