# Peer review of "Mixed Neuroendocrine Non-Neuroendocrine Neoplasms: A Systematic Review of a Controversial and Underestimated Diagnosis"

_jcm, 2020, doi:10.3390/jcm9010273_

Round 1

Reviewer 1 Report

This review presents the results of a thorough analysis of the literature on mixed neuroendocrine non-neuroendocrine neoplasm (MiNEN) of the GEP, gathering 91 studies after careful selection. Even though the conclusion is not decisive in terms of optimal treatment, it helps the clinician to better understand the pathogenesis, prognostic and disease by offering a concise view of the current literature.

Author Response

See attached ("Author's responses to Reviewers' comments").

Please also see new amended version of the manuscript (with minor tracked changes), new Figure 2 and amended Table 1.

Reviewer 2 Report

Overall, nicely performed and described systematic review. I have only a few comments:

Results:

Paragraph 1, suggest defining "CRs" and "RSs" to avoid confusion Line 181, the 92.5% does not match up with 258 of 571 patients; I assume it should be 528 Was palliative intent specifically documented? Or is that inferred by the patient having had Stage IV disease? This treatment paragraph is quite a lot to digest at once; I would suggest breaking this into sections for systemic chemotherapy versus radiation therapy or another way to improve flow of the summary of treatments Any information on where recurrence or progression developed? Rates of metastasectomy and impact on survival?

Discussion:

Line 274 I think the wording of "...ex-goblet cell carcinoids..." is a little off In paragraph 3, I would mention that perhaps some patients underwent surgery with debulking intent, which has rationale in highly selected patients with pure NET or adenocarcinoma, and that in pure NETs, some clinicians advocate resection of the primary tumor even in the presence of disseminated disease 

Author Response

See attachment below (it is uploaded as 'cover letter', but it indeed contains 'Author's responses to Reviewers' comments'), new version of the manuscript with minor tracked changes, new Figure 2 and amended Table 1.

Reviewer 3 Report

In this review, Frizziero et al. attempted to outline the epidemiological, clinical and pathological characteristics, prognosis, and genetic/epigenetic data of MiNEN from different sites of origin, in addition to the most commonly adopted treatment strategies.

Major concerns

In the Results, it would be easier for the readers to understand the percentage distribution with pie charts or other graphs. In Line 109 of Page 3, the authors mentioned about Figure 1, but there was no figure available on the manuscript. In Line 178 of Page 4, the authors mentioned about Tables 1-3, but there was no table available on the manuscript. In Line 232 of Page 5, the authors mentioned about Tables 4, but there was no table available on the manuscript.

Minor concerns

In Line 181 of Page 4, “...surgery (92.5%; n=258)” should be “...surgery (92.5%; n=528)”. In Line 205 of Page 5,”Among RS, 26 (=2176)…” should be “Among RS, 26 (n=2176)…”. In Table S2, the abbreviation GCC (goblet cell carcinoid) should be mentioned in the footnote. In Line 379 of Page 8, the duplicated Table S1 and related description should be corrected (? Table S2).

Author Response

See attachment below (it is uploaded as 'cover letter', but it indeed contains 'Author's responses to Reviewers' comments'), new version of the manuscript with minor tracked changes, new Figure 2 and amended Table 1.

This manuscript is a resubmission of an earlier submission. The following is a list of the peer review reports and author responses from that submission.

Round 1

Reviewer 1 Report

This review presents the results of a thorough analysis of the literature on mixed adenoneuroendocrine carcinoma MANEC, gathering 87 studies after careful selection. Even though the conclusion is not decisive in terms of optimal treatment, it helps the clinician to better understand the pathogenesis, prognostic and disease by offering a concise view of the current literature.

This paper should suffer some major modifications:

1.     I would suggest using a more concise and academic style, the same tense through all the review and checking for minor spelling errors.

2.     I would suggest adding the link from row 48 in the bibliography

3.     There are two paragraphs stating the same idea row 117-118, row 43

4.     You stated in the supplementary table 1 that you used a review of case reports, according to Fig1.- the eligibility criteria, the review of case reports should be excluded. Nevertheless, you used some case reports separately in your review (row 166-167: “one was a review of CRs. Three cases included in the latter are also presented in the current manuscript as individual CRs [35, 38, 39].”)

5.     In exclusion criteria in Fig 1 you stated that cases reported where one of the two components were not malignant, were excluded, why in supplementary table 2 you report adenoma 3,5%.

6.     The following tables 1a, b, c, d are exhaustive and difficult to follow; and the contained data was presented. I would suggest presenting in tables only data that has not been presented in the main body of the manuscript.

7.      An abbreviation list will ease the reading of the article.

Reviewer 2 Report

Frizziero et al. performed a systematic review about MiNEN, elucidating how high-quality evidence available in literature is poor.

Although there is a merit in this manuscript, it investigates several aspects of this disease ( management, genetics and survival) and for this reason it has not a clear focus, and is prolix in some paragraphs.

Some issues need to be modified, in order to make this draft suitable for publication:

- The review includes publications until June 2017, but an update including at least 2018 should be considered. This change would make the manuscript more appealing, as the WHO 2017 classification has introduced the term “MiNEN”. Furthermore inclusion of only retrospective studies, and exclusion of case reports should be adopted; this modification would also simplify Tables.

- According to the novel WHO classification, this disease has to be named as “MiNEN” (as explained at page 5, lines 145-155). The title and the text should be updated adopting this term, citing the WHO 2017-2019 classifications and explaining that the database search used the “MANEC definition” for papers published previous to the last classifications. Discussion section should also be modified accordingly.

- Relying on the recent WHO classifications, the possibility of an intermediate-grade MiNEN, combining a non-neuroendocrine carcinoma and a well-differentiated NET, has to be cited (differently from what reported at page 2, lines 61-62). The text has to be updated accordingly. 

- in chapter 2, page 2-3, the statistical method adopted to calculate survival rates reported in the results section and in table 2 should be explained.

Minor comments:

- In the abstract, line 10, correct as “little is still known”.

- Table S2: report absolute numbers besides percentages.